developmental biology, evolution, genetics

sex determination, sex ratio, *Niveoscincus*, GSD, TSD, XY

**Author for correspondence:**
Peta Hill
e-mail: peta.hill@utas.edu.au

# Sex reversal explains some, but not all, climate-mediated sex ratio variation within a viviparous reptile

Peta Hill[1], Geoffrey M. While[1], Christopher P. Burridge[1], Tariq Ezaz[2], Kirke L. Munch[1], Mary McVarish[1] and Erik Wapstra[1]

[1]Discipline of Biological Sciences, University of Tasmania, Private Bag 5, Hobart, Tasmania 7000, Australia
[2]Institute for Applied Ecology, University of Canberra, Bruce, Australian Capital Territory 2601, Australia

PH, 0000-0002-6190-6426; GMW, 0000-0001-8122-9322; CPB, 0000-0002-8185-6091;
TE, 0000-0003-4763-1347; KLM, 0000-0003-2929-6805; EW, 0000-0002-2050-8026

Evolutionary transitions in sex-determining systems have occurred frequently yet understanding how they occur remains a major challenge. In reptiles, transitions from genetic to temperature-dependent sex determination can occur if the gene products that determine sex evolve thermal sensitivity, resulting in sex-reversed individuals. However, evidence of sex reversal is limited to oviparous reptiles. Here we used thermal experiments to test whether sex reversal is responsible for differences in sex determination in a viviparous reptile, *Carinascincus ocellatus*, a species with XY sex chromosomes and population-specific sex ratio response to temperature. We show that sex reversal is occurring and that its frequency is related to temperature. Sex reversal was unidirectional (phenotypic males with XX genotype) and observed in both high- and low-elevation populations. We propose that XX-biased genotypic sex ratios could produce either male- or female-biased phenotypic sex ratios as observed in low-elevation *C. ocellatus* under variable rates of XX sex reversal. We discuss reasons why sex reversal may not influence sex ratios at high elevation. Our results suggest that the mechanism responsible for evolutionary transitions from genotypic to temperature-dependent sex determination is more complex than can be explained by a single process such as sex reversal.

## 1. Background

In sexually reproducing organisms, the sex of an individual is determined either by genes (genetic-dependent sex determination; GSD) or by the environment it experiences during embryonic development (mostly temperature-dependent sex determination, TSD) [1]. In several vertebrate systems, however, the sex encoded genetically can be over-ridden by the environment [2], termed GSD plus environmental effects (GSD + EE; [3]). Mechanistic models suggest that this occurs when the expression of a gene that determines sex is temperature-sensitive [4,5]. Biased sex ratios can result when the homogametic sex (XX females or ZZ males) does not attain the encoded phenotype because of the temperature sensitivity of the sex-determining gene product, despite having two copies of that gene. When this occurs, development is diverted down an alternate pathway [4], resulting in individuals with a sex genotype–phenotype mismatch. This outcome is known as 'sex reversal' [2,6] and has been documented in fish [7,8], amphibians [9] and oviparous reptiles [4,10,11]. In amphibians and fish, both the homogametic and heterogametic sex can develop into the opposite phenotype [12,13].

Observations of sex reversal can provide evidence for incipient transitions between sex-determining systems, for example from GSD to TSD, which become complete when one of the sex chromosomes is lost. This process can

be quite rapid [5,14]. For example, temperature-induced sex reversal of genotypically male (ZZ) *Pogona vitticeps* (Agamidae) to female, and successful mating between ZZ males and ZZ females, result in the production of exclusively ZZ offspring [10]. In laboratory cohorts of *P. vitticeps*, such temperature-induced sex reversal leads to the loss of the W chromosome in one generation. Offspring sex is then determined entirely by temperature interacting with the ZZ genotype, facilitating a temperature-mediated transition from GSD to TSD [10,15]. This mechanism has also been implicated in the loss of the W chromosome reported in some wild populations of *P. vitticeps* [10,16]. Sex reversal has also been observed in *Acritoscincus duperreyi* (Scincidae) [6,15] and has been inferred in lacertids, testudines and geckos [6]. The high frequency of evolutionary transitions between GSD and TSD among reptiles [17,18], in combination with the likely occurrence of sex reversal [4,5,15,19], suggests sex reversal may be a key mechanism operating at the time of a transition between modes of sex determination [6,10,20]. However, the extent to which sex reversal facilitates transitions in sex determination requires further assessment [21].

The viviparous spotted snow skink, *Carinascincus ocellatus* (Scincidae), provides an outstanding opportunity to address this. Field-based, laboratory and theoretical work on *C. ocellatus* has identified intra-specific divergence in sex determination [22], which is rarely observed in amniotes, although reported in amphibians [23] and fish [24,25]. Long-term field and laboratory data show that sex ratios in a high-elevation population do not deviate from parity irrespective of seasonal/developmental temperature [22,26]. By contrast, in a low-elevation population, sex ratios correlate with temperature [22]; sex ratios are female-biased in warm seasons/developmental conditions and male-biased in cool seasons/developmental conditions [26,27]. The population-specific sex ratio response to temperature observed in *C. ocellatus* in the wild has been replicated by manipulating maternal thermal opportunity in the laboratory [22,26–30]. Low-elevation females subjected to cooler short basking treatments (e.g. 4 h) produce male-biased cohorts and those subjected to warmer long basking exposures (e.g. 10 h) produce female-biased cohorts [29,30]. By contrast, high-elevation females subjected to the same thermal regimes produce sex parity cohorts [22,26–30]. An adaptive explanation for these patterns is that the production of males or females is favoured at different temperatures at low elevation because the concomitant variation in birth date has sex-specific fitness consequences. Warm developmental conditions, and thus early birth, favour females at low elevation because birthdate strongly predicts the onset of maturity and thus reproductive output for females, but not males [22]. The shorter reproductive season at high elevation together with the high seasonal temperature fluctuations mean there are no benefits of early birth to either sex, and therefore, sex ratios remain balanced.

While the functional significance of population-specific sex determination is well understood, the mechanisms underpinning these differences are still unclear. Despite differences in sex determination, both high- and low-elevation populations of *C. ocellatus* share sex-linked genetic sequences supporting XY heterogamety [31,32]. This suggests that the high-elevation population has GSD and the low-elevation population has GSD + EE. Of the mechanisms that could explain GSD + EE at low elevation, sex-specific mortality is unlikely because palpation of gravid females reveals the number of ovulated eggs is equal to the number of offspring [22,29], which leaves sex reversal as the primary candidate mechanism [6,21].

Here, we test whether temperature-induced sex reversal is the mechanism responsible for the differences in sex ratio responses to temperature in *C. ocellatus*, and hence the divergence of GSD and GSD + EE systems. This would suggest that a sex-determining gene product has evolved temperature sensitivity at low elevation. We predicted that sex reversal would occur in a manner consistent with long-term sex ratio responses to temperature in *C. ocellatus*. We predicted warmer temperatures would produce sex-reversed (XY) females and cooler temperatures would produce sex-reversed (XX) males at low elevation because both male and female sex ratio biases have been observed in this population. In addition, we also predicted that genotypic sex would always be concordant with phenotypic sex at high elevation because sex ratios do not deviate from parity in this population.

## 2. Material and methods

### (a) Study species and study sites

*Carinascincus ocellatus* is a small Tasmanian endemic skink. Reproduction is annual and follows a similar pattern at high and low elevation [33]. In spring 2018, 100 pregnant females were collected from a high elevation (41 60′ S, 146 44′ E; elevation 1050 m.a.s.l.) and a low-elevation (42 33′ S, 147 50′ E; elevation 50 m.a.s.l.) site shortly after ovulation (ovulation dates are 1st October and 15th October at low and high elevations, respectively [33]) and well prior to sexual differentiation of offspring [34].

We compared the phenotypic and genetic sex of offspring over a range of developmental temperatures using two complementary experimental protocols. Specifically, females were allocated to either a 'thermoregulation' or a 'no thermoregulation' experiment. In the 'thermoregulation' experiment, we manipulated the thermal environment by mimicking variation in basking opportunity experienced by females in the wild across the temperature extremes of the *C. ocellatus* range [28,30,35–37]. Females were held under either a long or a short basking treatment within which they were provided access to a basking lamp for 10 h and 4 h per day, respectively. This protocol has previously reproduced sex ratio responses in the wild: (i) male-biased under reduced basking, female-biased under extended basking in low-elevation *C. ocellatus* and (ii) parity in high-elevation *C. ocellatus* [22,29]. Basking lamps were placed over 200 × 300 × 100 mm terraria as per [36]. This created a thermal gradient within the terraria from approximately 20°C to 37°C during the day. Temperature dropped to approximately 10°C at night. Terraria were randomly repositioned within the room three times a week to avoid positional effects.

In the 'no thermoregulation' experiment, we mimicked sex ratio experiments undertaken on the eggs of oviparous species [4,11]. This protocol removes females' ability to behaviourally optimize developmental conditions and therefore isolates the physiological responses to temperature with respect to sex determination during offspring development [38]. Females were held individually in terraria (150 × 200 × 100 mm), placed in incubators and held under one of three experimental daytime temperatures (33.0°C, 29.5°C, 26.0°C; 8.00–16.00). Temperature was lowered to 10°C for the remaining 16 h of the 24 h period across all treatments to approximate ambient overnight temperatures. To avoid positional effects, females were randomly

shuffled within incubators three times a week, and treatments were rotated through the incubators fortnightly. Twenty females were assigned to each treatment ('no thermoregulation': 33.0°C, 29.5°C and 26.0°C; 'thermoregulation': 10 h and 4 h). All terraria were maintained under UV lighting and either LED strip lighting (no thermoregulation) or fluorescent tube lighting (thermoregulation) for 14 h light : 10 h dark. All females were supplied with water *ad libitum* and were offered mealworms and fruit supplemented with vitamins three times weekly. Towards the end of gestation, all terraria were checked for offspring. Litter success was high for both populations in all treatments except the high-elevation population held at 26.0°C, where only four females produced viable litters; the remaining females aborted their litter before the end of parturition. Selection in this population leads pregnant females to maintain high levels of basking in cool thermal conditions to maintain optimal developmental temperatures during gestation and avoid the negative effects of delayed birth [35,39,40]. Low litter success for this population in the 26.0°C treatment therefore reflects preclusion of basking in the 'no thermoregulation' experiment. Upon birth, all offspring were weighed and their gestation length was calculated from birth date relative to ovulation dates of each population. Offspring were transferred to terraria and kept under long bask conditions until release.

## (b) Phenotypic and genetic sexing and sample collection

Phenotypic sex was assessed via hemipene eversion. In many reptiles, delayed regression of hemipenes in females makes phenotypic sexing via hemipene eversion unreliable [41]. However, in *C. ocellatus*, long-term experimental and field collection has shown that this technique is reliable. Specifically, offspring sex agrees with adult sex when individuals are subsequently recaptured; offspring with hemipenes (sexed as males at birth) have not appeared as adult females in subsequent experiments or field seasons [22,26,27,29], which would be expected if delayed hemipene regression resulted in an incorrect assignment as male [21]. For this study, offspring phenotypic sex was determined at least twice: on average 14 and 26 days after birth. All phenotypic sexing was performed by the same investigator (E.W.) and was blind to treatment, population and without prior sexing information. If initial and subsequent phenotypic sex differed, offspring sex was determined again two weeks after the second sexing (2.3% of offspring: 'no thermoregulation' $n = 6$ low elevation; 'thermoregulation' $n = 3$ low elevation, 1 high elevation).

Tail tip samples were taken for genotypic sexing. Approximately 5 mm of tail was sampled and stored in ethanol. DNA extraction and genotyping were performed by Diversity Arrays Technology (https://www.diversityarrays.com [42]) using DArTcap targeted genotyping. Genetic sex was assigned using a suite of 45 single-nucleotide polymorphism loci exhibiting sex linkage in either or both sex-determining systems of *C. ocellatus* [31]. Where a mismatch occurred between phenotypic and genetic sex, the individual was deemed to be sex reversed.

## (c) Statistical analysis

Gestation length is tightly linked to the thermal conditions experienced by females and is thus a good proxy for the thermal developmental conditions experienced by embryos [26,30,36]. To confirm that our treatments affected offspring temperature-specific development rate, we fit a linear model (lm) for each experiment (thermoregulation, no thermoregulation) with log-transformed gestation length as the response variable and treatment and population and their interaction as fixed factors.

To test for differences in the frequency of sex reversal between treatments and populations, we fit a generalized linear mixed model (glmm) with binomial error distribution for each experiment using a type II Walds $\chi^2$-test with offspring sex status (i.e. whether genotypic and phenotypic sex matched) as the dependent variable, and treatment, population and their interaction as fixed factors. We included maternal identity as a random effect, given mean litter sizes of 2.4 and 3.1 at low and high elevation, respectively. We used gestation length as a proxy for temperatures to model the proportion of sex-reversed individuals as a function of developmental conditions. This allowed us to pool the 'thermoregulation' and 'no thermoregulation' experiments without knowledge of the actual temperatures achieved by basking females in the 'thermoregulation' experiment. We used a logistic regression model (glm) with a binomial distribution using a type II Walds $\chi^2$-test with mean gestation length and population as fixed factors. Finally, we analysed whether treatment phenotypic sex ratio and cohort genetic sex ratio deviated from parity using Pearson's chi-squared analysis on counts of male and female and XX and XY offspring. Analyses were conducted with R [43] using the 'lme4' (glmm, glm [44]), 'stats' (lm [43]) and 'car' (type II Wald's test [45]) packages.

## 3. Results

We identified sex-reversed *C. ocellatus* males (XX males) in both the 'no thermoregulation' and 'thermoregulation' experiments and in both the high- and low-elevation populations (table 1). No XY females were observed. We found XX and XY males and XX females within the same litter. Specifically, of the 19 and 10 litters containing XX males from low and high elevation, respectively, a total of five litters also contained XX females and XY males (two from low and three from high elevation). For both experiments, XX males were more commonly observed when thermal opportunity was decreased (table 1 and figure 1). These effects of developmental environment on sex reversal were consistent across both populations, i.e. there was no interaction between treatment and population (table 1).

We observed significant differences in gestation length between individuals under the different thermal conditions and between the two populations (figure 1 and table 2). These effects were consistent across both experiments (thermoregulation and no thermoregulation) and confirmed that gestation length is a robust proxy for developmental temperature in *C. ocellatus*. Specifically, individuals with lower thermal opportunities had longer gestations (figure 1). Gestation was also longer in individuals from the low-elevation population (figure 1). We found no significant interaction between treatment and population (table 2).

Across thermal treatments and populations, the proportion of XX male offspring was related to mean gestation length ($\chi_1^2 = 18.2$, $p < 0.01$; figure 2). This effect of gestation length was consistent across populations ($\chi_1^2 = 0.04$, $p = 0.86$) and there was no interaction between gestation length and population ($\chi_1^2 = 0.01$, $p = 0.91$).

Treatment phenotypic sex ratios ranged from 0.35 to 0.73 (table 3). Across both populations, male biases were more often observed in cooler or shorter treatments, where XX males occurred at a higher frequency, and female biases observed in the warmer and longer treatments, where fewer XX males were observed. The genetic sex ratio of offspring born during these experiments deviated significantly from

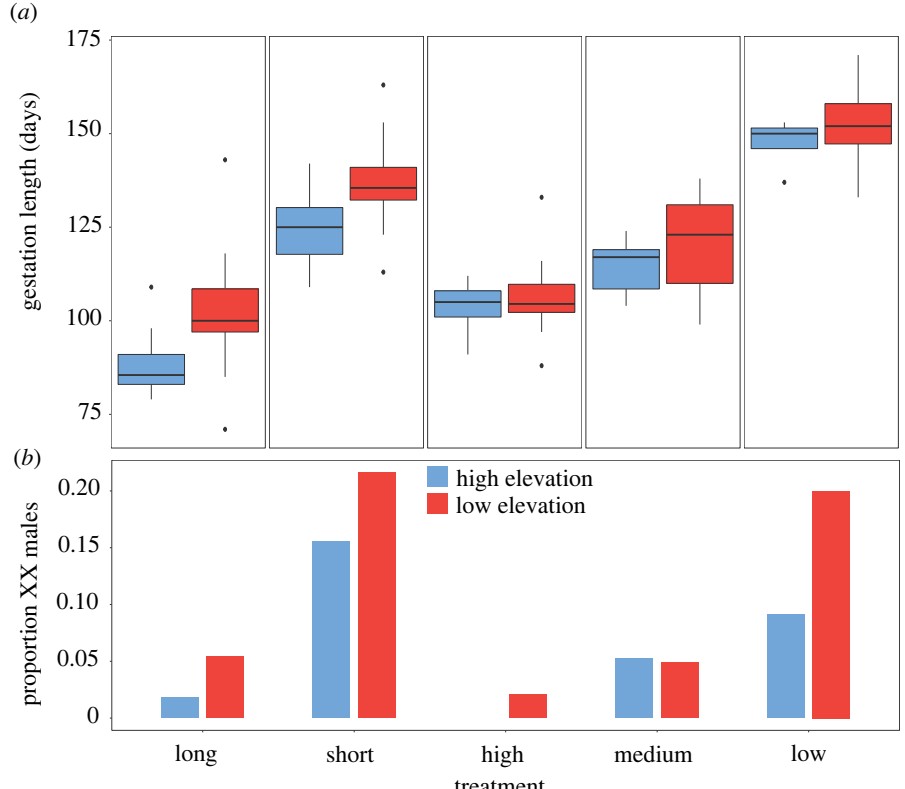

**Figure 1.** (*a*) Gestation lengths (days) and (*b*) proportion of sex-reversed male (XX male) *Carinascincus ocellatus* offspring for high-elevation (blue) and low-elevation (red) *C. ocellatus* females held in 'thermoregulation' (long, 10 h; short, 4 h) and 'no thermoregulation' (high, 33.0°C; medium, 29.5°C; low, 26.0°C) experiments. (Online version in colour.)

**Table 1.** Number of offspring from low- and high-elevation *Carinascincus ocellatus* females (20 per treatment) held in 'no thermoregulation' or 'thermoregulation' experiments, and the number that were sex reversed (XX males). Numbers in parentheses are total number of offspring born including mortalities (for which phenotypic sex could not be assessed) and proportion of XX males. Summary statistics from type II Wald's test of glmm assessing the effect of treatment and population on sex reversal are included.

| no thermoregulation experiment treatment: | low elevation | | high elevation | |
|---|---|---|---|---|
| | no. offspring (including mortalities) | no. XX males (proportion) | no. offspring (including mortalities) | no. XX males (proportion) |
| high (33.0°C) | 48 (48) | 1 (0.02) | 60 (60) | 0 (0.00) |
| med (29.5°C) | 41 (42) | 2 (0.05) | 57 (60) | 3 (0.05) |
| low (26.0°C) | 31 (35) | 6 (0.19) | 11 (12) | 1 (0.09) |
| treatment $\chi^2_2 = 9.9$, $p = {<}0.01$ | population $\chi^2_2 = 0.90$, $p = 0.34$ | | interaction $\chi^2_2 = 0.61$, $p = 0.74$ | |

| thermoregulation experiment treatment: | low elevation | | high elevation | |
|---|---|---|---|---|
| | no. offspring (including mortalities) | no. XX males (proportion) | no. offspring (including mortalities) | no. XX males (proportion) |
| long (10 h) | 37 (42) | 2 (0.05) | 56 (60) | 1 (0.02) |
| short (4 h) | 37 (43) | 8 (0.22) | 45 (47) | 7 (0.16) |
| treatment $\chi^2_1 = 7.7$, $p = {<}0.01$ | population $\chi^2_1 = 1.1$, $p = 0.29$ | | interaction $\chi^2_1 = 0.26$, $p = 0.61$ | |

parity in the low-elevation cohort, but not in the high-elevation cohort (table 4).

## 4. Discussion

We provide the first evidence that temperature-induced sex reversal occurs in viviparous reptiles. Rates of sex reversal in *C. ocellatus* varied with temperature, but contrary to our predictions, appear unidirectional: female genotype (XX) to male phenotype. Sex reversals occurred in both low- and high-elevation populations, despite previous work failing to identify sex ratio biases in the high-elevation populations in either field or laboratory experiments [22,26]. Therefore, while our results provide a key mechanism underpinning temperature-dependent sex determination in this system,

**Table 2.** Summary statistics from linear models testing the effect of treatment and population and their interaction on gestation length in high- and low-elevation populations of *Carinascincus ocellatus* with divergent sex determination.

| no thermoregulation experiment | | |
|---|---|---|
| treatment $F_{(2,1.65)} = 150.6$, $p < 0.01$ | population $F_{(1,0.02)} = 3.9$, $p = 0.05$ | interaction $F_{(2,0.005)} = 0.5$, $p = 0.63$ |
| thermoregulation experiment | | |
| treatment $F_{(1,1.83)} = 180.3$, $p < 0.01$ | population $F_{(1,0.25)} = 24.9$, $p < 0.01$ | interaction $F_{(1,0.014)} = 1.4$, $p = 0.23$ |

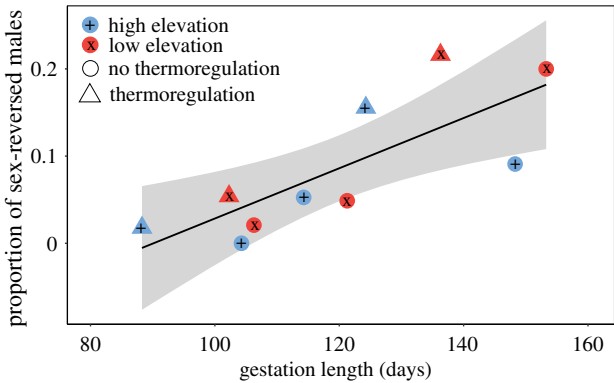

**Figure 2.** Relationship between mean gestation length (days) and the proportion of sex-reversed XX male offspring (number XX males/total number of offspring per treatment) from 'no thermoregulation' (circles) and 'thermoregulation' (triangles) experiments, and low-elevation (red, with crosses x) and high-elevation (blue, with plus signs +) *Carinascincus ocellatus* populations. (Online version in colour.)

they do not fully explain long-term population-specific sex ratio responses to temperature in *C. ocellatus*.

The low-elevation population of *C. ocellatus* exhibits substantial variation in sex ratios associated with developmental temperature [22,26,27,29]. Specifically, sex ratios are female-biased in warm seasons/developmental conditions and male-biased in cool seasons/developmental conditions and this has been linked to the sex-specific fitness benefits of birthdate [22]. Our observations of sex reversal provide an obvious mechanism by which such male-biased sex ratios can be achieved in cool conditions by reversal of the XX genotype to male phenotype [27,29]. However, they do not offer an immediate solution for the female-biased ratios in warm seasons. Female biases under warm developmental conditions could be achieved without XY sex reversal if the cohort genotypic sex ratio at fertilization is sufficiently XX-biased and the rate of XX reversal is low. Likewise, sex ratio parity could be achieved when an underlying XX genotypic bias is negated by the rate of XX sex reversals. Our results support this hypothesis because all the female offspring in our study have the XX genotype, including those from treatments with a slight female bias (e.g. long bask). Multiple paternity in this species [46] and other snow skinks [47] means that it is unlikely that entire litters will be fathered by an XX male, however, XX males mating in the population would explain the genetic sex ratios in favour of the XX genotype observed at low elevation. Excess females resulting from XX-biased genetic sex ratios rather than from the reversal of the XY genotype as we predicted, avoids the likely mortality of YY offspring from mating between XY females and XY males [48,49]. Hence, unidirectional sex reversal can produce bidirectional phenotypic biases because it can maintain a genotypic sex

bias that then facilitates male or female phenotypic biases based on the rate of sex reversal.

Sex reversals were also identified in our high-elevation population. Furthermore, sex reversal exhibited strong temperature sensitivity in our high-elevation population consistent with the patterns we observed in the low-elevation population. Combined with the fact that the two populations exhibit relatively consistent sex ratio responses (electronic supplementary material, figure S1), and those sex ratios responses are in the same direction as the sex reversals (more male-biased sex ratios in treatments with a greater proportion of sex reversals), this provides strong evidence that sex reversals are a key mechanism contributing to temperature-sensitive sex determination in this species. However, a key caveat to this is that extensive field and laboratory work has previously shown a lack of temperature effects on sex ratios at high elevation compared to a strong effect at low elevation [26,29]. How do we explain these seemingly contrasting results? One explanation is that sex reversals are always possible at high elevation, but they are rarely expressed because they are masked by female basking behaviour. High-elevation females show a greater propensity to bask both in the laboratory and in the wild than do low-elevation females [50]. This may allow females to compensate for the poor thermal conditions that greatly reduce, or even eliminate the production of XX male reversal at high elevation. While intuitively sensible, this cannot be the sole explanation given: (i) we typically observe consistent effects on other thermally sensitive traits (e.g. gestation length) in the high-elevation population even in the absence of thermal effects on sex ratio [26] and (ii) we observed thermal effects on sex reversals in high-elevation females in our thermoregulation treatment despite strong female compensation (P Hill, GM While, CP Burridge, T Ezaz, KL Munch, M McVarish, E Wapstra 2019, unpublished data). Clearly more work is needed to unpick the nature of the relationship between temperature, sex reversals and sex ratio variation if we are to explain how these effects play out across climatic regimes. One key way forward would be to extensively genotype cohorts from our long-term data on *C. ocellatus* for which we have known phenotypic sex ratios as this would show whether (and how much) sex reversal is occurring in the wild in both populations and allow tracking of the prevalence of XX males across seasons. These sample sets are much larger than the numbers achieved here and would therefore have greater power to tease apart the subtleties of these effects. Our observation of XX male offspring from both populations under experimental conditions suggests that the mechanism responsible for geographic divergence in sex ratio responses to climate in *C. ocellatus* is more complex than can be explained by a single process such as sex reversal

**Table 3.** Sex ratios of offspring cohorts from high- and low-elevation *Carinascincus ocellatus* females held in 'no thermoregulation' (high 33.0℃, med 29.5℃, low 26.0℃) or 'thermoregulation' (long 10 h, short 4 h) experiments during gestation. Summary statistics from Pearson's chi-squared analysis on counts of male and female offspring are included.

| population | treatment | offspring | sex ratio (M/M + F) | Pearson's $\chi^2$-test |
|---|---|---|---|---|
| high elevation | long (10 h) | 56 | 0.43 | $\chi_1^2 = 0.86, p = 0.35$ |
| | short (4 h) | 45 | 0.62 | $\chi_1^2 = 1.80, p = 0.18$ |
| | high (33.0℃) | 60 | 0.60 | $\chi_1^2 = 2.77, p = 0.10$ |
| | med (29.5℃) | 57 | 0.63 | $\chi_1^2 = 4.41, p = 0.04$ |
| | low (26.0℃) | 11 | 0.73 | $\chi_1^2 = 2.27, p = 0.13$ |
| low elevation | long (10 h) | 37 | 0.35 | $\chi_1^2 = 3.42, p = 0.06$ |
| | short (4 h) | 37 | 0.59 | $\chi_1^2 = 1.32, p = 0.25$ |
| | high (33.0℃) | 48 | 0.56 | $\chi_1^2 = 0.75, p = 0.39$ |
| | med (29.5℃) | 41 | 0.44 | $\chi_1^2 = 0.38, p = 0.54$ |
| | low (26.0℃) | 31 | 0.61 | $\chi_1^2 = 3.12, p = 0.08$ |

**Table 4.** Cohort genetic sex ratios for offspring from low- and high-elevation populations of *Carinascincus ocellatus* females held in 'thermoregulation' (long 10 h, short 4 h) and 'no thermoregulation' (high 33.0℃, med 29.5℃, low 26.0℃) experiments. Summary statistics from Pearson's $\chi^2$-test of the deviation of the genetic sex ratios from parity are included.

| population | male genotypes | | female genotypes | | genetic sex ratio | Pearson's $\chi^2$-test |
|---|---|---|---|---|---|---|
| | XY | XX | XY | XX | XY : XX | |
| low elevation | 80 | 19 | 0 | 94 | 80 : 113 | $\chi^2 = 5.6, p = 0.02$ |
| high elevation | 119 | 12 | 0 | 98 | 119 : 110 | $\chi^2 = 0.35, p = 0.55$ |

and that their divergence in sex determination is at an early stage of its evolution. This is consistent with evidence from population genetic data, which suggest that the high- and low-elevation populations of *C. ocellatus* diverged between 0.61 and 0.92 million years ago [51].

Sex reversals occur in the homogametic sex in *C. ocellatus*, which is consistent with the gene dosage model of reptile sex determination [4]. In this model, sexual phenotype is determined by the dosage of a sex-determining gene, as occurs in birds [51], rather than the presence or absence of a sex-determining gene, as occurs in therians [52]. Our results therefore suggest that the sex-determining gene in *C. ocellatus* is dosage dependent. When an X-linked sex-determining gene product in an XX/XY dosage system is sensitive to temperature, for example, through downregulation of gene transcription or denaturing of the resulting gene product, XX genotypes can fail to reach the threshold for female phenotype and are diverted down the male developmental pathway. These results are consistent with other lizards that have provided evidence supporting dosage-mediated sex determination in ZZ/ZW and XX/XY systems [4,19]. Y-specific genetic sequence has been identified in both *C. ocellatus* populations [31,32] and Y-specific and W-specific sequence has been identified in two other species where temperature interacts with sex chromosomes to determine sex (*Acritoscincus duperreyi* and *P. vitticeps*, respectively [19,53–55]). However, XX *C. ocellatus* and *A. duperreyi* can reverse to male phenotype, and ZZ *P. vitticeps* can reverse to female phenotype, raising questions around the function of the sex-specific chromosome in initiating sexual differentiation. Unlike mammals and birds where sex-determining

genes are highly conserved (*Sry* and *dmrt1*, respectively [1]), there are several candidate sex-determining genes among reptile groups [6,56–58]. Identifying the genes responsible for sex determination in reptiles with GSD + EE, mapping these to sex chromosomes and describing their function in sex concordant and sex-reversed individuals is a key step towards understanding the control of sexual phenotype in dosage-dependent sex-determining systems.

Population-specific sex ratio response to temperature suggests that *C. ocellatus* is undergoing an evolutionary transition in sex determination [22,26]. In addition to the fitness benefits of early birth to females at low elevation that appear to be driving this divergence [22], such a transition will also be influenced by gene flow and heritable variation in the threshold for sex reversal [59]. Gene flow of GSD genotypes into a population can inhibit its transition to TSD [59]. While gene flow between high- and low-elevation *C. ocellatus* populations is negligible [60,61], GSD genotypes could enter the low-elevation population from more proximate sources and thus attenuate a transition to TSD. Heritable variation in the threshold for sex reversal also decreases the likelihood of a transition to TSD because the maintenance of variation in the sex reversal threshold across generations will favour the persistence of a mixed system of sex determination [59]. In *C. ocellatus*, litters with XX males also contained XX females, therefore, there is within-litter variation in thresholds for sexual phenotype. In addition, although the frequency of sex reversal is positively associated with decreasing developmental temperature, sex reversal in warmer/longer treatments is evidence that some XX offspring possess a higher threshold for development as female and are more likely to reverse to

male. This could be linked to multiple paternity in *C. ocellatus* because XX offspring of sex-reversed males may be more sensitive to temperature and therefore more likely to reverse, as has been reported for *P. vitticeps* [10]. Wider surveys of sex determination systems in *C. ocellatus* in addition to quantifying the threshold for sexual phenotype and understanding gene flow across its range will expose further characteristics of the transition in sex determination underway in this system. Our study adds to the growing body of work investigating the dynamic nature of the evolution of sex determination systems across reptiles and highlights the need to combine observations from wild populations and laboratory experiments to fully appreciate the complexity of the mechanisms involved in transitions.

Ethics. All work was approved by the University of Tasmania Animal Ethics Committee, ethics approval no. A0017544.

Data accessibility. The datasets supporting this article have been uploaded as part of the electronic supplementary material.

The data are provided in the electronic supplementary material [62].

Authors' contributions. P.H.: conceptualization, data curation, formal analysis, funding acquisition, methodology, project administration and writing—original draft; G.M.W.: conceptualization, formal analysis, methodology and writing—review and editing; C.P.B.: conceptualization, formal analysis, funding acquisition, project administration, supervision and writing—review and editing; T.E.: conceptualization, funding acquisition, project administration, supervision and writing—review and editing; K.L.M.: formal analysis, investigation and writing—review and editing; M.M.: formal analysis, investigation and writing—review and editing; E.W.: conceptualization, formal analysis, funding acquisition, methodology, project administration, supervision, validation and writing—review and editing.

All authors gave final approval for publication and agreed to be held accountable for the work performed therein.

Conflict of interest declaration. We declare we have no conflict of interest to declare.

Funding. This study was funded by Australia and Pacific Science Foundation (grant no. APSF 17/8) and Holsworth Wildlife Research Endowment.

Acknowledgements. We thank L. Fitzpatrick, D. Merry, E. Rohrlach and S. Sengupta for field collection and valuable contributions and discussions.

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
