## [Peer Review File · Proceedings of the Royal Society B: Biological Sciences]

Review History

RSPB-2021-1778.R0 (Original submission)

Review form: Reviewer 1 (Daniel Jeffries)

Recommendation

Major revision is needed (please make suggestions in comments)

Scientific importance: Is the manuscript an original and important contribution to its field?

Acceptable

General interest: Is the paper of sufficient general interest?

Good

Quality of the paper: Is the overall quality of the paper suitable?

Acceptable

Is the length of the paper justified?

Yes

Should the paper be seen by a specialist statistical reviewer?

No

Do you have any concerns about statistical analyses in this paper? If so, please specify them explicitly in your report.

No

It is a condition of publication that authors make their supporting data, code and materials available - either as supplementary material or hosted in an external repository. Please rate, if applicable, the supporting data on the following criteria.

Is it accessible?

Yes

Is it clear?

Yes

Is it adequate?

Yes

Do you have any ethical concerns with this paper?

No

Comments to the Author

In this study, Hill et al raise *C. ocellatus* offspring under several differing temperature treatments in order to test the hypothesis that patterns of sex ratio distortions which correlate with temperature in the wild are the result of temperature induced sex reversals. To do this they phenotypically and genetically sex offspring and compare the number of sex reversals among treatments and populations. The results show mixed findings, there are more sex reversals in the low altitude population as expected, however it is difficult to tell from the way the results are currently presented whether this is sufficient to explain the higher sex ratio distortion in this population in the wild. Further, and against expectations, a non-trivial number of sex reversals were observed in experiments involving individuals from the high altitude population. The authors conclude, and I support them in doing so, that things are more complicated than first thought, and either things other than sex reversals are at work, or the dynamics of sex reversals are complex.

In my opinion, this is a solid study. The experiments carried out and the statistics performed were appropriate for testing the main hypothesis of this paper and I believe that the conclusions drawn are reasonable. I would like to see genotype proportions in the results if possible, and a more detailed description of how the experimental results relate to field observations. And the way the paper is presented does need some work to improve clarity and objectivity, these I have pointed out specifically below.

Kind regards.

Dr. Dan Jeffries

Line 70. I propose removing "From an evolutionary perspective"

Line 71-74. I'm a bit sceptical about the latter part of this sentence ("whose sex is determined entirely by temperature), perhaps it needs some explanation or clarification. I understand that if a sex reversed XX male mates with an XX female, all offspring would be XX. But I find it hard to believe that all offspring then immediately have TSD. That would suggest an extremely high (if not 100%) heritability of sex reversal from day one. Isn't this more likely to be gradual? I.e. sex reversed parents might tend to have more sex reversed offspring than non-sex reversed parents, and this proportion increases and TSD spreads through the pop as genetic TSD mechanisms evolve / fix?

Line 75-84 - I find the reasoning of this section (which underpins the whole study really), a bit misleading. I agree with the authors when they say "[the co-occurrence of sex reversals and GSD/TSD transitions is] strongly suggestive of an evolutionary link between these two traits". But as it reads now, it seems you are suggesting that temperature induced sex reversal plays a causal role in transitions to TSD (e.g. L. 79: "the extent to which it underpins transitions between modes of sex determination". This is surely circular. Instead, can we not view this link as one between a process (the transition) and a mechanism (sex reversals)? i.e. temperature induced sex reversal is just the mechanism by which a GSD - TSD transition occurs. Or even that sex reversals are just a necessary consequence of the transition?

L. 108. It would seem clearer here to say that this study tests whether temperature induced sex reversals are the mechanism by which sex ratio distortion occurs in the lower altitude populations (rather than sex specific mortality).

L. 135 - 138. The authors mention here that it has already been shown that manipulating basking opportunity causes skews in sex ratio. So I am only now realising that the main contribution of the present paper is to confirm that the cause of this skew is indeed sex reversal. However this is not clear enough in the intro of the paper. So it would seem appropriate to more explicitly discuss these previous studies in the introduction, highlighting the work that has already been done to study this, and the other potential causes of this skew. At the moment, this is only addressed by a single sentence on L. 105 - 107.

L. 145. Do the authors mean that this isolates the physiological responses from basking behaviour response specifically? Perhaps they could be a bit clearer about what each of these protocols adds and why its important to separate these things?

L. 237-240. This sentence structure is quite hard to follow, please rephrase. Also, I think it is misleading to refer to the numbers in Table 3 in terms of odds. E.g. the authors state that it is 21.5 times more likely that an individual will be sex reversed in a low temp treatment than in a high temp treatment. While technically true, in reality that is a difference of 5 sex reversed offspring (i.e. 6/31 in the low treatment vs. 1/48 in the high treatment). Given these are such small numbers, I don't think odds comparisons are appropriate. Consider if the experiment was repeated and the results for Low and High treatments were 5/31 and 2/48, that would drastically change these odds. I think it would be more representative to simply state the proportions of offspring that were sex reversed.

L. 258 - I am not sure why gestation length vs sex reversal is this an informative comparison, or one worthy of a main text figure? Aren't these things just both correlates of temp? Plotting this in a figure implies (to me at least), that the authors expect some causal link between them.

L. 263 - It is confusing to give sex ratios in two forms (50:50, 0.36-0.72) in the same sentence.

L. 270 - this table would benefit from the inclusion of the number of individuals in each treatment in Table 4.

L. 293 - given that the authors found only one type of sex reversal, I think it would be simpler throughout the discussion to refer to the results in the context of XX males rather than "sex reversed offspring".

L. 296 - 298. Again, I am unclear on the importance of the correlation between gestation length and sex reversal. The authors refer to gestation length as a "proxy for developmental temperature", but here we exactly know the developmental temperature so no proxy is needed. Is this a useful proxy for field observations perhaps? Clarification of this in the methods or even in the intro would be useful.

L. 298 - 301. The authors say (on line 291) that their results do not fully explain long term field observations, and they find a good amount of sex reversal in the high altitude population, where sex ratios are typically 50:50. Yet on line 300 they say that sex reversals play a pivotal role in explaining sex ratio distortions. These sentences are at somewhat at odds with each other, so it would be nice here to have a more in-depth and objective discussion how sex reversal frequencies in their experiments match up to sex ratio distortions observed in the wild and how important they really may be. For example, lets say 10% of XX offspring turn out male in the experiments here, assuming we started from 50:50 XX:XY embryos, that would result in a 40:60 female to male sex ratio. Is that close to observed sex ratios in seasons similar to those represented by the temperature experiments. If not, how much variation might it explain? How “pivotal” is it?

L. 315 - 317. To say that “[thermosensitive GSD] has been modified in one [population] relative to the other by selection” would now seem like a strong assertion now given that the main result of this paper is seemingly that sex reversals do not explain all of the variation in sex ratios across temperatures treatments. And also given that in *A. duperreyi* sex reversals occur in the same way as here but wild populations have more sex reversal at high altitudes, this would suggest that other important factors are at play in addition to sex reversals alone.

L. 385-387. I suppose the authors must have data on the XX/XY genotype ratios in their offspring datasets? That would be very useful to include! Indeed, if, as the authors speculate on lines 328 - 333, it is an X dosage system, and XX individuals lend themselves to sex reversal more readily than XY individuals, then having XX biased genotypes at birth would result in a more “flexible” cohort and I could imagine this being advantageous.

Review form: Reviewer 2

Recommendation

Major revision is needed (please make suggestions in comments)

Scientific importance: Is the manuscript an original and important contribution to its field?

Good

General interest: Is the paper of sufficient general interest?

Acceptable

Quality of the paper: Is the overall quality of the paper suitable?

Acceptable

Is the length of the paper justified?

Yes

Should the paper be seen by a specialist statistical reviewer?

No

Do you have any concerns about statistical analyses in this paper? If so, please specify them explicitly in your report.

No

It is a condition of publication that authors make their supporting data, code and materials available - either as supplementary material or hosted in an external repository. Please rate, if applicable, the supporting data on the following criteria.

Is it accessible?

Yes

Is it clear?

Yes

Is it adequate?

Yes

Do you have any ethical concerns with this paper?

No

Comments to the Author

The authors examine how temperature affects sex ratio in both a high elevation and low elevation population of *Carinascincus ocellatus* using two different temperature control protocols. They find that regardless of population of origin or protocol of temperature control, the population sex ratios of pregnant females kept in cooler treatments were more likely to have sex-reversed males among the offspring than pregnant females kept in warmer treatments. This is in contrast to natural populations, where the higher elevation population generally has equal sex ratios, which has been attributed to more strict GSD. Furthermore, they find sex-reversal in only one direction: XX males, but not XY females.

I found the paper interesting. As the title suggests, the results are a bit difficult to interpret, and I appreciate the authors openly stating so. I only have a few suggestions/comments/questions, which I hope will help the authors improve the manuscript.

1. Do females at higher altitudes have more opportunities for basking? If yes, this seems relevant for explaining how at even cooler ambient temperatures the higher elevation population maintains equal sex ratios.
2. Lines 25-27. If under experimental conditions, the authors cannot show that the two population produce different sex ratios under the same temperature treatments, is it really fair to say that this explains some of the difference between the low and high elevation populations? Or perhaps the lower elevation is cooler (which seems counterintuitive)?
3. Lines 105-107. The author simply state that sex-specific mortality is unlikely to explain sex ratio biases in the lower elevation population. Perhaps state why explicitly?
4. Is it safe to assume that males father complete clutches (many of my comments assume this)? Or is it possible to have multiple fathers per clutch?
5. Lines 24-25. The intraspecific variation in sex ratio is referring to GSD and ESD or two different GSDs? I assumed ESD and GSD the way it is written in the abstract, but in lines 184-186 it sounds like this species has 2 different GSDs? I honestly can't tell for sure. This needs to be made more explicit, and if two different GSDs, then more detail on both sex-determining systems and their distribution would be helpful.
6. Lines 373-402. The authors describe here how an XX male mating with an XX female could produce both male- and female-biased sex ratios in the absence of sex-reversed (XY) females. It seems to me that this could go a long way to explaining their results. Do the authors find XY males and XX males in the same clutch? Because if not, then wouldn't this suggest the father of the clutch is an XX male? More generally, it seems like including some information on the clutch-level response to temperature would shed some light on the findings presented.

Decision letter (RSPB-2021-1778.R0)

20-Sep-2021

Dear Dr Hill:

I am writing to inform you that your manuscript RSPB-2021-1778 entitled "Sex reversal explains some, but not all, climate mediated sex ratio variation within a viviparous reptile" has, in its current form, been rejected for publication in Proceedings B.

This action has been taken on the advice of referees, who have recommended that substantial revisions are necessary. With this in mind we would be happy to consider a resubmission, provided the comments of the referees are fully addressed. However please note that this is not a provisional acceptance.

Sincerely,
Dr Maurine Neiman
mailto: proceedingsb@royalsociety.org

Associate Editor
Board Member: 1
Comments to Author:

This manuscript has now been reviewed by two experts and myself. We all agree that this is an interesting study that makes an important contribution to the literature. Here, the authors test how temperature affects sex ratio in both a high elevation and low elevation population of the spotted snow skink to investigate dynamics of sex determination in wild populations.

Both reviewers have clearly thought deeply about the study and raised important questions that should be addressed.

Notably:

Both reviewers pointed out that the link between the experimental results and natural patterns in the wild could be improved. I agree, and would urge the authors to clarify this.

Currently, the authors frame the paper to investigate how sex reversals play a causal role in transitions between sex determination mechanisms. The authors should clarify this - do they mean that sex reversals are just a necessary consequences of the transition or are they suggesting something else.

Previous work should be discussed in more detail and the advance that this research represents highlighted more clearly.

Please report the proportions of offspring that were sex reversed rather than just the ratios.

Reviewer(s)' Comments to Author:

Referee: 1

Comments to the Author(s)

In this study, Hill et al raise *C. ocellatus* offspring under several differing temperature treatments in order to test the hypothesis that patterns of sex ratio distortions which correlate with temperature in the wild are the result of temperature induced sex reversals. To do this they phenotypically and genetically sex offspring and compare the number of sex reversals among treatments and populations. The results show mixed findings, there are more sex reversals in the low altitude population as expected, however it is difficult to tell from the way the results are currently presented whether this is sufficient to explain the higher sex ratio distortion in this population in the wild. Further, and against expectations, a non-trivial number of sex reversals were observed in experiments involving individuals from the high altitude population. The authors conclude, and I support them in doing so, that things are more complicated than first thought, and either things other than sex reversals are at work, or the dynamics of sex reversals are complex.

In my opinion, this is a solid study. The experiments carried out and the statistics performed were appropriate for testing the main hypothesis of this paper and I believe that the conclusions drawn are reasonable. I would like to see genotype proportions in the results if possible, and a more detailed description of how the experimental results relate to field observations. And the way the paper is presented does need some work to improve clarity and objectivity, these I have pointed out specifically below.

Kind regards.

Dr. Dan Jeffries

Line 70. I propose removing "From an evolutionary perspective"

Line 71-74. I'm a bit sceptical about the latter part of this sentence ("whose sex is determined entirely by temperature), perhaps it needs some explanation or clarification. I understand that if a sex reversed XX male mates with an XX female, all offspring would be XX. But I find it hard to believe that all offspring then immediately have TSD. That would suggest an extremely high (if not 100%) heritability of sex reversal from day one. Isn't this more likely to be gradual? I.e. sex reversed parents might tend to have more sex reversed offspring than non-sex reversed parents, and this proportion increases and TSD spreads through the pop as genetic TSD mechanisms evolve / fix?

Line 75-84 - I find the reasoning of this section (which underpins the whole study really), a bit misleading. I agree with the authors when they say "[the co-occurrence of sex reversals and GSD/TSD transitions is] strongly suggestive of an evolutionary link between these two traits". But as it reads now, it seems you are suggesting that temperature induced sex reversal plays a causal role in transitions to TSD (e.g. L. 79: "the extent to which it underpins transitions between modes of sex determination". This is surely circular. Instead, can we not view this link as one between a process (the transition) and a mechanism (sex reversals)? i.e. temperature induced sex reversal is just the mechanism by which a GSD - TSD transition occurs. Or even that sex reversals are just a necessary consequence of the transition?

L. 108. It would seem clearer here to say that this study tests whether temperature induced sex reversals are the mechanism by which sex ratio distortion occurs in the lower altitude populations (rather than sex specific mortality).

L. 135 - 138. The authors mention here that it has already been shown that manipulating basking opportunity causes skews in sex ratio. So I am only now realising that the main contribution of the present paper is to confirm that the cause of this skew is indeed sex reversal. However this is not clear enough in the intro of the paper. So it would seem appropriate to more explicitly discuss these previous studies in the introduction, highlighting the work that has already been done to study this, and the other potential causes of this skew. At the moment, this is only addressed by a single sentence on L. 105 - 107.

L. 145. Do the authors mean that this isolates the physiological responses from basking behaviour response specifically? Perhaps they could be a bit clearer about what each of these protocols adds and why its important to separate these things?

L. 237-240. This sentence structure is quite hard to follow, please rephrase. Also, I think it is misleading to refer to the numbers in Table 3 in terms of odds. E.g. the authors state that it is 21.5 times more likely that an individual will be sex reversed in a low temp treatment than in a high temp treatment. While technically true, in reality that is a difference of 5 sex reversed offspring (i.e. 6/31 in the low treatment vs. 1/48 in the high treatment). Given these are such small numbers, I don't think odds comparisons are appropriate. Consider if the experiment was repeated and the results for Low and High treatments were 5/31 and 2/48, that would drastically change these odds. I think it would be more representative to simply state the proportions of offspring that were sex reversed.

L. 258 - I am not sure why gestation length vs sex reversal is this an informative comparison, or one worthy of a main text figure? Aren't these things just both correlates of temp? Plotting this in a figure implies (to me at least), that the authors expect some causal link between them.

L. 263 - It is confusing to give sex ratios in two forms (50:50, 0.36-0.72) in the same sentence.

L. 270 - this table would benefit from the inclusion of the number of individuals in each treatment in Table 4.

L. 293 - given that the authors found only one type of sex reversal, I think it would be simpler throughout the discussion to refer to the results in the context of XX males rather than "sex reversed offspring".

L. 296 - 298. Again, I am unclear on the importance of the correlation between gestation length and sex reversal. The authors refer to gestation length as a "proxy for developmental temperature", but here we exactly know the developmental temperature so no proxy is needed. Is this a useful proxy for field observations perhaps? Clarification of this in the methods or even in the intro would be useful.

L. 298 - 301. The authors say (on line 291) that their results do not fully explain long term field observations, and they find a good amount of sex reversal in the high altitude population, where sex ratios are typically 50:50. Yet on line 300 they say that sex reversals play a pivotal role in explaining sex ratio distortions. These sentences are at somewhat at odds with each other, so it would be nice here to have a more in-depth and objective discussion how sex reversal frequencies in their experiments match up to sex ratio distortions observed in the wild and how important they really may be. For example, lets say 10% of XX offspring turn out male in the experiments here, assuming we started from 50:50 XX:XY embryos, that would result in a 40:60 female to male sex ratio. Is that close to observed sex ratios in seasons similar to those represented by the temperature experiments. If not, how much variation might it explain? How "pivotal" is it?

L. 315 - 317. To say that “[thermosensitive GSD] has been modified in one [population] relative to the other by selection” would now seem like a strong assertion now given that the main result of this paper is seemingly that sex reversals do not explain all of the variation in sex ratios across temperatures treatments. And also given that in *A. duperreyi* sex reversals occur in the same way as here but wild populations have more sex reversal at high altitudes, this would suggest that other important factors are at play in addition to sex reversals alone.

L. 385-387. I suppose the authors must have data on the XX/XY genotype ratios in their offspring datasets? That would be very useful to include! Indeed, if, as the authors speculate on lines 328 - 333, it is an X dosage system, and XX individuals lend themselves to sex reversal more readily than XY individuals, then having XX biased genotypes at birth would result in a more “flexible” cohort and I could imagine this being advantageous.

Referee: 2

Comments to the Author(s)

The authors examine how temperature affects sex ratio in both a high elevation and low elevation population of *Carinascincus ocellatus* using two different temperature control protocols. They find that regardless of population of origin or protocol of temperature control, the population sex ratios of pregnant females kept in cooler treatments were more likely to have sex-reversed males among the offspring than pregnant females kept in warmer treatments. This is in contrast to natural populations, where the higher elevation population generally has equal sex ratios, which has been attributed to more strict GSD. Furthermore, they find sex-reversal in only one direction: XX males, but not XY females.

I found the paper interesting. As the title suggests, the results are a bit difficult to interpret, and I appreciate the authors openly stating so. I only have a few suggestions/comments/questions, which I hope will help the authors improve the manuscript.

1. Do females at higher altitudes have more opportunities for basking? If yes, this seems relevant for explaining how at even cooler ambient temperatures the higher elevation population maintains equal sex ratios.
2. Lines 25-27. If under experimental conditions, the authors cannot show that the two population produce different sex ratios under the same temperature treatments, is it really fair to say that this explains some of the difference between the low and high elevation populations? Or perhaps the lower elevation is cooler (which seems counterintuitive)?
3. Lines 105-107. The author simply state that sex-specific mortality is unlikely to explain sex ratio biases in the lower elevation population. Perhaps state why explicitly?
4. Is it safe to assume that males father complete clutches (many of my comments assume this)? Or is it possible to have multiple fathers per clutch?
5. Lines 24-25. The intraspecific variation in sex ratio is referring to GSD and ESD or two different GSDs? I assumed ESD and GSD the way it is written in the abstract, but in lines 184-186 it sounds like this species has 2 different GSDs? I honestly can't tell for sure. This needs to be made more explicit, and if two different GSDs, then more detail on both sex-determining systems and their distribution would be helpful.
6. Lines 373-402. The authors describe here how an XX male mating with an XX female could produce both male- and female-biased sex ratios in the absence of sex-reversed (XY) females. It seems to me that this could go a long way to explaining their results. Do the authors find XY males and XX males in the same clutch? Because if not, then wouldn't this suggest the father of the clutch is an XX male? More generally, it seems like including some information on the clutch-level response to temperature would shed some light on the findings presented.

Author's Response to Decision Letter for (RSPB-2021-1778.R0)

See Appendix A.

RSPB-2022-0689.R0

Review form: Reviewer 3

Recommendation

Accept as is

Scientific importance: Is the manuscript an original and important contribution to its field?

Excellent

General interest: Is the paper of sufficient general interest?

Excellent

Quality of the paper: Is the overall quality of the paper suitable?

Excellent

Is the length of the paper justified?

Yes

Should the paper be seen by a specialist statistical reviewer?

No

Do you have any concerns about statistical analyses in this paper? If so, please specify them explicitly in your report.

No

It is a condition of publication that authors make their supporting data, code and materials available - either as supplementary material or hosted in an external repository. Please rate, if applicable, the supporting data on the following criteria.

Is it accessible?

Yes

Is it clear?

Yes

Is it adequate?

Yes

Do you have any ethical concerns with this paper?

No

Comments to the Author

I believe the authors have sufficiently addressed all the comments by the reviewers and incorporated many of the suggestions into the manuscript. I do not have any concerns, and I believe the manuscript would be a great contribution to the journal.

Note: in Table 4, the numbers for the XX genotypes do not add up to 114, but to 113. Comparing the numbers with the previous tables, I suspect that there is a typo in the number of XX females (should possibly be 95 instead of 94).

Decision letter (RSPB-2022-0689.R0)

09-May-2022

Dear Dr Hill

I am pleased to inform you that your manuscript RSPB-2022-0689 entitled "Sex reversal explains some, but not all, climate mediated sex ratio variation within a viviparous reptile" has been accepted for publication in Proceedings B.

The referee(s) have recommended publication, but also suggest a minor revision to your manuscript. Therefore, I invite you to respond to the referee(s)' comments and revise your manuscript. Because the schedule for publication is very tight, it is a condition of publication that you submit the revised version of your manuscript within 7 days. If you do not think you will be able to meet this date please let us know.

When submitting your revision please upload a file under "**Response to Referees**" - in the "File Upload" section. This should document, point by point, how you have responded to the reviewers' and Editors' comments, and the adjustments you have made to the manuscript. We also require a **copy of the revised manuscript showing track changes** to be uploaded.

4) Data accessibility section and data citation

It is a condition of publication that data supporting your paper are made available either in the electronic supplementary material. Authors must complete the 'data accessibility' section in the

submission system. This should list the database and accession number for all data from the article that has been made publicly available, for instance:

NB. From April 1 2013, peer reviewed articles based on research funded wholly or partly by RCUK must include, if applicable, a statement on how the underlying research materials – such as data, samples or models – can be accessed.

[http://datadryad.org/submit?journalID=RSPB&manu=\(Document not available\)](http://datadryad.org/submit?journalID=RSPB&manu=(Document%20not%20available)) which will take you to your unique entry in the Dryad repository. If you have already submitted your data to dryad you can make any necessary revisions to your dataset by following the above link.

Please include the **Dryad DOI in the Data Accessibility section** and reference in the paper's bibliography.

Please see our Data Sharing Policies (<https://royalsociety.org/journals/authors/author-guidelines/>).

6) A media summary: a short non-technical summary (up to 100 words) of the key findings/importance of your manuscript.

Once again, thank you for submitting your manuscript to **Proceedings B** and I look forward to receiving your revision. If you have any questions at all, please do not hesitate to get in touch.

Sincerely,

Dr Maurine Neiman

Associate Editor

Board Member

Comments to Author:

The authors have thoughtfully addressed the reviewers comments in detail and the manuscript is much improved as a result. I am happy to recommend this paper for acceptance.

Reviewer(s)' Comments to Author:

Referee: 3

Comments to the Author(s).

I believe the authors have sufficiently addressed all the comments by the reviewers and incorporated many of the suggestions into the manuscript. I do not have any concerns, and I believe the manuscript would be a great contribution to the journal.

Note: in Table 4, the numbers for the XX genotypes do not add up to 114, but to 113. Comparing the numbers with the previous tables, I suspect that there is a typo in the number of XX females (should possibly be 95 instead of 94).

Author's Response to Decision Letter for (RSPB-2022-0689.R0)

See Appendix B.

Decision letter (RSPB-2022-0689.R1)

12-May-2022

Dear Dr Hill

I am pleased to inform you that your manuscript entitled "Sex reversal explains some, but not all, climate mediated sex ratio variation within a viviparous reptile" has been accepted for publication in Proceedings B.

Your article has been estimated as being 9 pages long. Our Production Office will be able to confirm the exact length at proof stage.

Data Accessibility section

Open Access

Paper charges

You are allowed to post any version of your manuscript on a personal website, repository or preprint server. However, the work remains under media embargo and you should not discuss it

with the press until the date of publication. Please visit <https://royalsociety.org/journals/ethics-policies/media-embargo> for more information.

Sincerely,
Proceedings B
<mailto:proceedingsb@royalsociety.org>

Appendix A

9th April 2022

Dear Dr Neiman,

We would like to express our appreciation to yourself and the two reviewers for taking the time to review and comment on our manuscript “Sex reversal explains some, but not all, climate mediated sex ratio variation within a viviparous reptile.” We have revised our manuscript in line with these comments and hope this revised version is ready for publication in *Proceedings of the Royal Society B*.

Our responses are embedded in the text below, (highlighted in this colour) and address each suggestion.

Best regards,

Peta Hill on behalf of all co-authors.

Associate Editor

Board Member: 1

Comments to Author:

This manuscript has now been reviewed by two experts and myself. We all agree that this is an interesting study that makes an important contribution to the literature. Here, the authors test how temperature affects sex ratio in both a high elevation and low elevation population of the spotted snow skink to investigate dynamics of sex determination in wild populations.

Both reviewers have clearly thought deeply about the study and raised important questions that should be addressed.

Notably:

Both reviewers pointed out that the link between the experimental results and natural patterns in the wild could be improved. I agree, and would urge the authors to clarify this.

Currently, the authors frame the paper to investigate how sex reversals play a causal role in transitions between sex determination mechanisms. The authors should clarify this - do they mean that sex reversals are just a necessary consequences of the transition or are they suggesting something else.

Previous work should be discussed in more detail and the advance that this research represents highlighted more clearly.

Please report the proportions of offspring that were sex reversed rather than just the ratios.

We thank the Associate Editor and both reviewers for their time and effort in assessing our manuscript. We have made the changes (outlined below) in response to this feedback and feel this process has improved our manuscript. We have addressed reviewer feedback and the manuscript now includes genetic sex ratios from each population and clutch level sexual phenotype and genotype results. In addition, we have clarified our interpretations in places and have improved the links between previous work, our experimental results and natural patterns in the wild. We provide more in-depth details of previous work on this system and

clearly discuss the advances we have made to current understanding of transitions in sex determination. We have improved the results section by stating our most significant result first. We include a list of citations used in our responses to reviewers at the end of this document.

Reviewer(s)' Comments to Author:

Referee: 1

Comments to the Author(s)

In this study, Hill et al raise *C. ocellatus* offspring under several differing temperature treatments in order to test the hypothesis that patterns of sex ratio distortions which correlate with temperature in the wild are the result of temperature induced sex reversals. To do this they phenotypically and genetically sex offspring and compare the number of sex reversals among treatments and populations. The results show mixed findings, there are more sex reversals in the low altitude population as expected, however it is difficult to tell from the way the results are currently presented whether this is sufficient to explain the higher sex ratio distortion in this population in the wild. Further, and against expectations, a non-trivial number of sex reversals were observed in experiments involving individuals from the high altitude population. The authors conclude, and I support them in doing so, that things are more complicated than first thought, and either things other than sex reversals are at work, or the dynamics of sex reversals are complex.

In my opinion, this is a solid study. The experiments carried out and the statistics performed were appropriate for testing the main hypothesis of this paper and I believe that the conclusions drawn are reasonable. I would like to see genotype proportions in the results if possible, and a more detailed description of how the experimental results relate to field observations. And the way the paper is presented does need some work to improve clarity and objectivity, these I have pointed out specifically below.

Kind regards.

Dr. Dan Jeffries

We thank Referee 1, Dr Jeffries for their time and effort in assessing our study and we are grateful for the thoughtful and constructive comments. We have taken on Dr Jeffries' suggestions and have made changes to our manuscript as outlined below.

Line 70. I propose removing "From an evolutionary perspective"

Response: We agree that this phrasing could be improved and have changed it accordingly. Now L52 reads: "Observations of sex reversal provide evidence for incipient transitions between sex determining systems, such as between GSD to TSD, which become complete when one of the sex chromosomes is lost, for example producing single-sex populations."

Line 71-74. I'm a bit sceptical about the latter part of this sentence ("whose sex is determined entirely by temperature), perhaps it needs some explanation or clarification. I understand that

if a sex reversed XX male mates with an XX female, all offspring would be XX. But I find it hard to believe that all offspring then immediately have TSD. That would suggest an extremely high (if not 100%) heritability of sex reversal from day one. Isn't this more likely to be gradual? I.e. sex reversed parents might tend to have more sex reversed offspring than non-sex reversed parents, and this proportion increases and TSD spreads through the pop as genetic TSD mechanisms evolve / fix?

Response: Thank you for raising this point. The XX offspring resulting from a mating between an XX male and XX female have lost the sex-specific chromosome and therefore their sex can only be determined via mechanisms involving the homogametic chromosome. With no temperature override, XX offspring are all female or, when temperature has induced sex reversal in the parent, we assume offspring have inherited the threshold for sex reversal from parents and that (dependent on temperature) some XX offspring will develop as males. There is evidence for this from *Pogona vitticeps* (Central bearded dragon) including evidence that offspring from sex reversed parents are reversed more frequently [1] and we now support our assertion by providing an example of the temperature-mediated loss of a sex chromosome in *P. vitticeps* with the inclusion and modification as follows:

From L55: “For example, temperature-induced sex reversal of genotypically male (ZZ) *Pogona vitticeps* (Agamidae) to female, and successful mating between ZZ males and ZZ females, results in the production of exclusively ZZ offspring [10]. In laboratory cohorts of *P. vitticeps*, such temperature-induced sex reversal leads to the loss of the W chromosome in one generation. Offspring sex is then determined entirely by temperature interacting with the ZZ genotype, facilitating a temperature mediated transition from GSD to TSD [10, 15].”

Line 75-84 - I find the reasoning of this section (which underpins the whole study really), a bit misleading. I agree with the authors when they say “[the co-occurrence of sex reversals and GSD/TSD transitions is] strongly suggestive of an evolutionary link between these two traits”. But as it reads now, it seems you are suggesting that temperature induced sex reversal plays a causal role in transitions to TSD (e.g. L. 79: “the extent to which it underpins transitions between modes of sex determination”. This is surely circular. Instead, can we not view this link as one between a process (the transition) and a mechanism (sex reversals)? i.e. temperature induced sex reversal is just the mechanism by which a GSD - TSD transition occurs. Or even that sex reversals are just a necessary consequence of the transition?

Response: We believe there is evidence for a causal link between temperature induced sex reversal and evolutionary transitions in sex determination. This putative link has been addressed in previous literature [2-5] and there is both theoretical [2] and empirical data [1, 5] that has led to these conclusions. *C. ocellatus* provides an opportunity to test this in a system undergoing an incipient transition in sex determination. We have modified the wording to support our reasoning with more citations as follows:

L62 “Sex reversal has also been observed in *Acritoscincus duperreyi* (Scincidae) [6, 15] and has been inferred in lacertids, testudines and geckos [6]. The high frequency of evolutionary transitions between GSD and TSD among reptiles [17, 18], in combination with the likely occurrence of sex reversal [4, 5, 15, 19], suggests sex reversal may be a key mechanism operating at the time of a transition between modes of sex determination [6, 10, 20]. However, the extent to which sex reversal facilitates transitions in sex determination requires further assessment [21].”

L. 108. It would seem clearer here to say that this study tests whether temperature induced sex reversals are the mechanism by which sex ratio distortion occurs in the lower altitude populations (rather than sex specific mortality).

Response: Thank you for this advice, we have clarified our aim by changing the text now at L99 to read: “Here, we test whether temperature-induced sex reversal is the mechanism responsible for the differences in sex ratio responses to temperature in *C. ocellatus*, and hence divergence of GSD and GSD+EE systems.”

L. 135 - 138. The authors mention here that it has already been shown that manipulating basking opportunity causes skews in sex ratio. So I am only now realising that the main contribution of the present paper is to confirm that the cause of this skew is indeed sex reversal. However, this is not clear enough in the intro of the paper. So it would seem appropriate to more explicitly discuss these previous studies in the introduction, highlighting the work that has already been done to study this, and the other potential causes of this skew. At the moment, this is only addressed by a single sentence on L. 105 - 107.

Response: Thank you for highlighting this omission; we agree that further coverage of the background to this study is necessary. We have now added an extra paragraph in the introduction that covers further relevant background for this system.

L69 to L90: “The viviparous spotted snow skink, *Carinascincus ocellatus* (Scincidae) provides an outstanding opportunity to explore the potential role that sex reversal plays in transitions in sex determination. Field-based, laboratory and theoretical work on *C. ocellatus* has identified intra-specific divergence in sex determination [22] which is rarely observed in amniotes, although reported in amphibians [23] and fish [24, 25]. Long-term field and laboratory data show that sex ratios in a high elevation population do not deviate from parity irrespective of seasonal/developmental temperature [22, 26]. In contrast, in a low elevation population sex ratios correlate with temperature [22]; sex ratios are female biased in warm seasons/developmental conditions and male biased in cool seasons/developmental conditions [26, 27]. The population-specific sex ratio response to temperature observed in *C. ocellatus* in the wild has been replicated by manipulating maternal thermal opportunity in the laboratory [22, 26-30]. Low elevation females subjected to cooler short basking treatments (e.g., 4 hours) produce male biased cohorts and those subjected to warmer long basking exposures (e.g., 10 hours) produce female biased cohorts [29, 30]. In contrast, high elevation females subjected to the same thermal regimes produce sex parity cohorts [22, 26-30]. An adaptive explanation for these patterns is that the production of males or females is favoured at different temperatures at low elevation because the concomitant variation in birth date has sex-specific fitness consequences. Warm developmental conditions, and thus early birth, favour females at low elevation because birthdate strongly predicts the onset of maturity and thus reproductive output for females, but not males [22]. The shorter reproductive season at high elevation together with the high seasonal temperature fluctuations mean there are no benefits of early birth to either sex, and therefore, sex ratios remain balanced.”

L. 145. Do the authors mean that this isolates the physiological responses from basking behaviour response specifically? Perhaps they could be a bit clearer about what each of these protocols adds and why its important to separate these things?

Response: Our experiments represent two different ways of manipulating temperature - one, that we typically use (for example, [6]), whereby we manipulate temperature indirectly and allow females to choose their preferred basking temperature during gestation and another where we manipulate temperature directly, comparable to similar studies using egg laying species where offspring develop at set temperatures (for example, [1]). The latter is important in this context because it removes maternal influence on developmental temperature and thus gives us a potentially greater ability to isolate physiological process related to specific temperatures. L158 now reads “This protocol removes female’s ability to behaviourally optimise developmental conditions and therefore isolates the physiological responses to temperature with respect to sex determination during offspring development [38].”

L. 237-240. This sentence structure is quite hard to follow, please rephrase. Also, I think it is misleading to refer to the numbers in Table 3 in terms of odds. E.g. the authors state that it is 21.5 times more likely that an individual will be sex reversed in a low temp treatment than in a high temp treatment. While technically true, in reality that is a difference of 5 sex reversed offspring (i.e. 6/31 in the low treatment vs. 1/48 in the high treatment). Given these are such small numbers, I don’t think odds comparisons are appropriate. Consider if the experiment was repeated and the results for Low and High treatments were 5/31 and 2/48, that would drastically change these odds. I think it would be more representative to simply state the proportions of offspring that were sex reversed.

Response: We thank reviewer 1 for pointing this out. We have now included the number and proportion of sex reversed offspring in table 1 and removed the use of odds from the results section.

L205 of the results now reads: “For both experiments, sex reversed individuals were more commonly observed when thermal opportunity was decreased (Table 1, Figure 1). These effects of developmental environment on sex reversal were consistent across both populations, i.e., there was no interaction between treatment and population (Table 1).”

Table 1. Number of offspring from low elevation and high elevation *Carinascincus ocellatus* females (20 per treatment) held in ‘no thermoregulation’ or ‘thermoregulation’ experiments, and number that were sex reversed (XX males). Numbers in parentheses are total number of offspring born including mortalities (for which phenotypic sex could not be assessed) and proportion of XX males. Summary statistics from type II Wald’s test of GLMM assessing the effect of treatment and population on sex reversal are included.

No thermoregulation experiment	Low elevation		High elevation		
	No. offspring (including mortalities)	No. XX males (proportion)	No. offspring (including mortalities)	No. XX males (proportion)	
Treatment:					
High (33.0°C)	48 (48)	1 (0.02)	60 (60)	0 (0.00)	
Med (29.5°C)	41 (42)	2 (0.05)	57 (60)	3 (0.05)	
Low (26.0°C)	31 (35)	6 (0.19)	11 (12)	1 (0.09)	
Treatment $\chi^2_{(3)} = 9.9$, p = <0.01		Population $\chi^2_{(2)} = 0.90$, p = 0.34		Interaction $\chi^2_{(2)} = 0.61$, p = 0.74	
Thermoregulation Experiment	Low elevation		High elevation		
	No. offspring (including mortalities)	No. XX males (proportion)	No. offspring (including mortalities)	No. XX males (proportion)	
Treatment:					
Long (10 h)	37 (42)	2 (0.05)	56 (60)	1 (0.02)	
Short (4 h)	37 (43)	8 (0.22)	45 (47)	7 (0.16)	
Treatment $\chi^2_{(1)} = 7.7$, p = <0.01		Population $\chi^2_{(1)} = 1.1$, p = 0.29		Interaction $\chi^2_{(1)} = 0.26$, p = 0.61	

L. 258 - I am not sure why gestation length vs sex reversal is this an informative comparison, or one worthy of a main text figure? Aren’t these things just both correlates of temp? Plotting this in a figure implies (to me at least), that the authors expect some causal link between them.

Response: This is a valid question. This figure is included because gestation length serves to indicate the thermal conditions experienced by offspring and therefore allows us to make comparisons between experiment types (thermoregulation and no thermoregulation) without knowledge of the actual body temperature achieved by females in the thermoregulation experiment. We think that this figure illustrates nicely how strongly sex reversals are related to temperature whether females bask or not. We have clarified this at L178 (methods section) with: “Gestation length is tightly linked to the thermal conditions experienced by females and is thus a good proxy for the thermal developmental conditions experienced by embryos [26, 30, 36].”

And at L189: “We used gestation length as a proxy for temperatures to model the proportion of sex reversed individuals as a function of developmental conditions. This allowed us to pool the ‘thermoregulation’ and ‘no thermoregulation’ experiments without knowledge of the actual temperatures achieved by basking females in the ‘thermoregulation’ experiment.”

L. 263 - It is confusing to give sex ratios in two forms (50:50, 0.36-0.72) in the same sentence.

Response: Thank you for highlighting this, we agree this is confusing and have amended this to read at L253 “Treatment phenotypic sex ratios ranged from 0.35 to 0.73 (Table 3).”

L. 270 - this table would benefit from the inclusion of the number of individuals in each treatment in Table 4.

Response: We agree with reviewer 1’s assessment and have amended Tale 4, now at L260 as follows:

Table 3. Sex ratios of offspring cohorts from high and low elevation *Carinascincus ocellatus* females held in ‘no thermoregulation’ or ‘thermoregulation’ experiments during gestation. Summary statistics from Pearson’s chi-squared analysis on counts of male and female offspring are included.

Population	Treatment	Offspring	Sex ratio (M/M+F)	Pearson’s χ^2 test
High elevation	Long (10h)	56	0.43	$\chi^2 = 0.86, p = 0.35$
	Short (4h)	45	0.62	$\chi^2 = 1.80, p = 0.18$
	High (33.0°C)	60	0.60	$\chi^2 = 2.77, p = 0.10$
	Med (29.5°C)	57	0.63	$\chi^2 = 4.41, p = 0.04$
	Low (26.0°C)	11	0.73	$\chi^2 = 2.27, p = 0.13$
Low elevation	Long (10h)	37	0.35	$\chi^2 = 3.42, p = 0.06$
	Short (4h)	37	0.59	$\chi^2 = 1.32, p = 0.25$
	High (33.0°C)	48	0.56	$\chi^2 = 0.75, p = 0.39$
	Med (29.5°C)	41	0.44	$\chi^2 = 0.38, p = 0.43$
	Low (26.0°C)	31	0.61	$\chi^2 = 3.12, p = 0.08$

L. 293 - given that the authors found only one type of sex reversal, I think it would be simpler throughout the discussion to refer to the results in the context of XX males rather than “sex reversed offspring”.

Response: We agree that using the term ‘XX males’ is simpler and clearer and have made the appropriate changes at L203, L205, L253, L255, L288, L 299, L302, L347

L. 296 - 298. Again, I am unclear on the importance of the correlation between gestation length and sex reversal. The authors refer to gestation length as a “proxy for developmental temperature”, but here we exactly know the developmental temperature, so no proxy is needed. Is this a useful proxy for field observations perhaps? Clarification of this in the methods or even in the intro would be useful.

Response: We appreciate this comment and acknowledge that we did not fully explain our reasoning behind using gestation length as a proxy for developmental temperature in the

manuscript. Given that we only know the temperature experienced by gravid females in our ‘no thermoregulation’ experiment, the use of gestation length as a proxy for developmental temperature allows us to make comparisons between experiment types (thermoregulation and no thermoregulation) without knowledge of the actual body temperature achieved by females in the thermoregulation experiment. We have clarified this at L178 (methods section) with: “Gestation length is tightly linked to the thermal conditions experienced by females and is thus a good proxy for the thermal developmental conditions experienced by embryos [26, 30, 36].”

And at L189: “We used gestation length as a proxy for temperatures to model the proportion of sex reversed individuals as a function of developmental conditions. This allowed us to pool the ‘thermoregulation’ and ‘no thermoregulation’ experiments without knowledge of the actual temperatures achieved by basking females in the ‘thermoregulation’ experiment.”

L. 298 - 301. The authors say (on line 291) that their results do not fully explain long term field observations, and they find a good amount of sex reversal in the high altitude population, where sex ratios are typically 50:50. Yet on line 300 they say that sex reversals play a pivotal role in explaining sex ratio distortions. These sentences are at somewhat at odds with each other, so it would be nice here to have a more in-depth and objective discussion how sex reversal frequencies in their experiments match up to sex ratio distortions observed in the wild and how important they really may be. For example, lets say 10% of XX offspring turn out male in the experiments here, assuming we started from 50:50 XX:XY embryos, that would result in a 40:60 female to male sex ratio. Is that close to observed sex ratios in seasons similar to those represented by the temperature experiments. If not, how much variation might it explain? How “pivotal” is it?

Response: we agree that this assertion was too strong and have tempered our interpretation. We now provide a more in-depth discussion of the disparity between our observations of sex reversal in both high and low elevation populations, and sex ratio biases only occurring in the low elevation population. Beginning at L274 of the discussion:

“The low elevation population of *C. ocellatus* exhibits substantial variation in sex ratios associated with developmental temperature [22, 27, 29]. Specifically, sex ratios are female biased in warm seasons/developmental conditions and male biased in cool seasons/developmental conditions and this has been linked to the sex-specific fitness benefits of birthdate [22]. Our observations of sex reversal provide an obvious mechanism by which such male biased sex ratios can be achieved in cool seasons by reversal of the XX genotype to male phenotype [22, 26], but do not offer an immediate solution for the female biased ratios in warm seasons. However, female biases under warm developmental conditions could be achieved without XY sex reversal if the cohort genotypic sex ratio at fertilization is sufficiently XX biased and the rate of XX reversal is low. Likewise, sex ratio parity could be achieved when an underlying XX genotypic bias is negated by the rate of XX sex reversals. Our results support this hypothesis because all the female offspring in our study have the XX genotype, including those from treatments with a slight female bias (e.g., long bask). Multiple paternity in this species [46] and other snow skinks [47] means that it is unlikely that entire litters will be fathered by an XX male, however, XX males mating in the population would explain the genetic sex ratios in favour of the XX genotype observed at low elevation. Excess females resulting from XX biased genetic sex ratios rather than from the reversal of the XY genotype as we predicted, avoids the likely mortality of YY offspring from mating between XY females and XY males [48, 49]. Hence, unidirectional sex reversal can produce

bidirectional phenotypic biases because it can maintain a genotypic sex bias that then facilitates male or female phenotypic biases based on the rate of sex reversal.

Sex reversals were also identified in our high elevation population. Furthermore, sex reversal exhibited strong temperature sensitivity in our high elevation population consistent with the patterns we observed in the low elevation population. Combined with the fact that the two populations exhibit relatively consistent sex ratio responses (Supplementary Figure 1), and those sex ratios responses are in the same direction as the sex reversals (more male biased sex ratios in treatments with a greater proportion of sex reversals), this provides strong support that sex reversals are a key mechanism contributing to temperature sensitive sex determination in this species. However, a key caveat to this is that extensive field and laboratory work has previously shown a lack of temperature effects on sex ratios at high elevation compared to a strong effect at low elevation [26, 29]. How do we explain these seemingly contrasting results? One explanation is that sex reversals are always possible at high elevation, but they are rarely expressed because they are masked by female basking behaviour. High elevation females show a greater propensity to bask both in the laboratory and in the wild than do low elevation females [50]. This may allow females to compensate for the poor thermal conditions which greatly reduce, or even eliminate the production of XX male reversal at high elevation. While intuitively sensible this cannot be the sole explanation given, i) we typically observe consistent effects on other thermally sensitive traits (e.g., gestation length) in the high elevation population even in the absence of thermal effects on sex ratio [26] and ii) we observed thermal effects on sex reversals in high elevation females in our thermoregulation treatment despite strong female compensation (Hill et al. unpublished data). Clearly more work is needed to unpick the nature of the relationship between temperature, sex reversals and sex ratio variation if we are to explain how these effects play out across climatic regimes. One key way forward would be to extensively genotype cohorts from our long-term data on *C. ocellatus* for which we have known phenotypic sex ratios as this would show whether (and how much) sex reversal is occurring in the wild in both populations and allow tracking of the prevalence of XX males across seasons. These sample sets are much larger than the numbers achieved here and would therefore have greater power to tease apart the subtleties of these effects. Our observation of XX male offspring from both populations under experimental conditions suggests that the mechanism responsible for geographic divergence in sex ratio responses to climate in *C. ocellatus* is more complex than can be explained by a single process such as sex reversal and that their divergence in sex determination is at an early stage of its evolution. This is consistent with evidence from population genetic data which suggests that the high and low elevation populations of *C. ocellatus* diverged between 0.61 and 0.92 million years ago [51].”

L. 315 - 317. To say that “[thermosensitive GSD] has been modified in one [population] relative to the other by selection” would now seem like a strong assertion now given that the main result of this paper is seemingly that sex reversals do not explain all of the variation in sex ratios across temperatures treatments. And also given that in *A. duperreyi* sex reversals occur in the same way as here but wild populations have more sex reversal at high altitudes, this would suggest that other important factors are at play in addition to sex reversals alone. Response: We agree with Reviewer 1’s comments and we have tempered this statement by deleting the phrasing used above and rewording this paragraph to include some discussion of other factors potentially at play. Now beginning at L318 “Our observation of XX male offspring from both populations under experimental conditions suggests that the mechanism responsible for geographic divergence in sex ratio responses to climate in *C. ocellatus* is more complex than can be explained by a single process such as sex reversal and that their

divergence in sex determination is at an early stage of its evolution. This is consistent with evidence from population genetic data which suggests that the high and low elevation populations of *C. ocellatus* diverged between 0.61 and 0.92 million years ago [50].”

L. 385-387. I suppose the authors must have data on the XX/XY genotype ratios in their offspring datasets? That would be very useful to include! Indeed, if, as the authors speculate on lines 328 - 333, it is an X dosage system, and XX individuals lend themselves to sex reversal more readily than XY individuals, then having XX biased genotypes at birth would result in a more “flexible” cohort and I could imagine this being advantageous.

Response: Thank you for highlighting the importance of the genetic sex ratio to this study. We do indeed have this information and have now included it as suggested. We now include the genetic sex ratios from each population’s cohort and report relevant litter characteristics in our results at L202: “In addition, we found XX and XY males and XX females within the same litter. Specifically, of the 19 and 10 litters containing XX males from low and high elevation respectively, a total of five litters also contained XX females and XY males (two from low and three from high elevation respectively).”

And

L244: “The genetic sex ratio of offspring born during these experiments deviated significantly from parity in the low elevation cohort, but not the high elevation cohort (Table 4).”

Table 4. Cohort genetic sex ratios for offspring from low and high elevation populations of *Carinascincus ocellatus* females held in ‘thermoregulation’ (Long 10h, Short 4h) and ‘no thermoregulation’ (High 33.0 °C, Med 29.5 °C, Low 26.0 °C) experiments. Summary statistics from Pearson’s χ^2 test of the deviation of the genetic sex ratios from parity are included.

Population	Male genotypes		Female genotypes		Genetic sex ratio	Pearson’s χ^2 test
	XY	XX	XY	XX	XY:XX	
Low elevation	80	19	0	94	80:114	$\chi^2 = 6.00, p = 0.01$
High elevation	119	12	0	98	119:110	$\chi^2 = 0.35, p = 0.55$

We discuss these findings at L281: “However, female biases under warm developmental conditions could be achieved without XY sex reversal if the cohort genotypic sex ratio at fertilization is sufficiently XX biased and the rate of XX reversal is low. Likewise, sex ratio parity could be achieved when an underlying XX genotypic bias is negated by the rate of XX sex reversals. Our results support this hypothesis because all the female offspring in our study have the XX genotype, including those from treatments with a slight female bias (e.g., long bask). Multiple paternity in this species [46] and other snow skinks [47] means that it is unlikely that entire litters will be fathered by an XX male, however, XX males mating in the population would explain the genetic sex ratios in favour of the XX genotype observed at low elevation. Excess females resulting from XX biased genetic sex ratios rather than from the reversal of the XY genotype as we predicted, avoids the likely mortality of YY offspring from mating between XY females and XY males [48, 49]. Hence, unidirectional sex reversal can produce bidirectional phenotypic biases because it can maintain a genotypic sex bias that then facilitates male or female phenotypic biases based on the rate of sex reversal.”

Referee: 2

Comments to the Author(s)

The authors examine how temperature affects sex ratio in both a high elevation and low elevation population of *Carinascincus ocellatus* using two different temperature control protocols. They find that regardless of population of origin or protocol of temperature control, the population sex ratios of pregnant females kept in cooler treatments were more likely to have sex-reversed males among the offspring than pregnant females kept in warmer treatments. This is in contrast to natural populations, where the higher elevation population generally has equal sex ratios, which has been attributed to more strict GSD. Furthermore, they find sex-reversal in only one direction: XX males, but not XY females.

I found the paper interesting. As the title suggests, the results are a bit difficult to interpret, and I appreciate the authors openly stating so. I only have a few suggestions/comments/questions, which I hope will help the authors improve the manuscript.

We thank Reviewer 2 for their time and effort in assessing our study and are grateful for the constructive comments. We have taken on many of Reviewer 2's suggestions and accordingly have made changes to our manuscript as outlined below. We feel this manuscript is now much improved and stronger as a result of Reviewer 2's feedback.

1. Do females at higher altitudes have more opportunities for basking? If yes, this seems relevant for explaining how at even cooler ambient temperatures the higher elevation population maintains equal sex ratios.

Response: This is a relevant question and one which has been investigated in this system. From previous work, we know that gravid *C. ocellatus* females from high elevation bask more than those from low elevation [7, 8] and this might explain how the high elevation population maintains equal sex ratios. However, in this experiment, in the treatments where females were free to choose how long they basked, even though high altitude females basked more than low altitude females we still saw sex reversal in both populations, suggesting other mechanisms also work to balance the sex ratio at high elevation. We have discussed this in more depth in this revised version of the manuscript.

For example, at L291: "Sex reversals were also identified in our high elevation population. Furthermore, sex reversal exhibited strong temperature sensitivity in our high elevation population consistent with the patterns we observed in the low elevation population. Combined with the fact that the two populations exhibit relatively consistent sex ratio responses (Supplementary Figure 1), and those sex ratios responses are in the same direction as the sex reversals (more male biased sex ratios in treatments with a greater proportion of sex reversals), this provides strong support that sex reversals are a key mechanism contributing to temperature sensitive sex determination in this species. However, a key caveat to this is that extensive field and laboratory work has previously shown a lack of temperature effects on sex ratios at high elevation compared to a strong effect at low elevation [26, 29]. How do we explain these seemingly contrasting results? One explanation is that sex reversals are always possible at high elevation, but they are rarely expressed because they are masked by female basking behaviour. High elevation females show a greater propensity to bask both in the laboratory and in the wild than do low elevation females [50]. This may allow females

to compensate for the poor thermal conditions which greatly reduce, or even eliminate the production of XX male reversal at high elevation. While intuitively sensible this cannot be the sole explanation given, i) we typically observe consistent effects on other thermally sensitive traits (e.g., gestation length) in the high elevation population even in the absence of thermal effects on sex ratio [26] and ii) we observed thermal effects on sex reversals in high elevation females in our thermoregulation treatment despite strong female compensation (Hill et al. unpublished data). Clearly more work is needed to unpick the nature of the relationship between temperature, sex reversals and sex ratio variation if we are to explain how these effects play out across climatic regimes. One key way forward would be to extensively genotype cohorts from our long-term data on *C. ocellatus* for which we have known phenotypic sex ratios as this would show whether (and how much) sex reversal is occurring in the wild in both populations and allow tracking of the prevalence of XX males across seasons. These sample sets are much larger than the numbers achieved here and would therefore have greater power to tease apart the subtleties of these effects. Our observation of XX male offspring from both populations under experimental conditions suggests that the mechanism responsible for geographic divergence in sex ratio responses to climate in *C. ocellatus* is more complex than can be explained by a single process such as sex reversal and that their divergence in sex determination is at an early stage of its evolution. This is consistent with evidence from population genetic data which suggests that the high and low elevation populations of *C. ocellatus* diverged between 0.61 and 0.92 million years ago [51].”

2. Lines 25-27. If under experimental conditions, the authors cannot show that the two population produce different sex ratios under the same temperature treatments, is it really fair to say that this explains some of the difference between the low and high elevation populations? Or perhaps the lower elevation is cooler (which seems counterintuitive)?

Response: Thank you for raising this point. The lower elevation population experiences warmer temperatures across the season than the high elevation population. High elevation females bask more opportunistically to counteract this [7, 8]. Both in the wild [9] and in previous experiments using the same basking regime we used here [6, 10], sex ratios are biased at low elevation but not at high, therefore, sex reversal at high elevation was unexpected. We believe sex reversal explains sex ratio biases at low elevation, hence the use of the word “some”, however, we have clarified our interpretation throughout the manuscript, including acknowledging that sex ratios remaining at parity at high elevation despite sex reversal requires further exploration. L26-29 in the abstract now reads: “We propose that XX biased genotypic sex ratios could produce either male or female biased phenotypic sex ratios as observed in low elevation *C. ocellatus* under variable rates of XX sex reversal. We also discuss reasons why sex reversal may not influence sex ratios at high elevation.”

Also, we have included as per the above, a more in-depth discussion of our findings at L270: “The low elevation population of *C. ocellatus* exhibits substantial variation in sex ratios associated with developmental temperature [22, 27, 29]. Specifically, sex ratios are female biased in warm seasons/developmental conditions and male biased in cool seasons/developmental conditions and this has been linked to the sex-specific fitness benefits of birthdate [22]. Our observations of sex reversal provide an obvious mechanism by which such male biased sex ratios can be achieved in cool seasons by reversal of the XX genotype to male phenotype [22, 26], but do not offer an immediate solution for the female biased ratios in warm seasons. However, female biases under warm developmental conditions could be achieved without XY sex reversal if the cohort genotypic sex ratio at fertilization is sufficiently XX biased and the rate of XX reversal is low. Likewise, sex ratio parity could be

achieved when an underlying XX genotypic bias is negated by the rate of XX sex reversals. Our results support this hypothesis because all the female offspring in our study have the XX genotype, including those from treatments with a slight female bias (e.g., long bask). Multiple paternity in this species [46] and other snow skinks [47] means that it is unlikely that entire litters will be fathered by an XX male, however, XX males mating in the population would explain the genetic sex ratios in favour of the XX genotype observed at low elevation. Excess females resulting from XX biased genetic sex ratios rather than from the reversal of the XY genotype as we predicted, avoids the likely mortality of YY offspring from mating between XY females and XY males [48, 49]. Hence, unidirectional sex reversal can produce bidirectional phenotypic biases because it can maintain a genotypic sex bias that then facilitates male or female phenotypic biases based on the rate of sex reversal.

Sex reversals were also identified in our high elevation population. Furthermore, sex reversal exhibited strong temperature sensitivity in our high elevation population consistent with the patterns we observed in the low elevation population. Combined with the fact that the two populations exhibit relatively consistent sex ratio responses (Supplementary Figure 1), and those sex ratios responses are in the same direction as the sex reversals (more male biased sex ratios in treatments with a greater proportion of sex reversals), this provides strong support that sex reversals are a key mechanism contributing to temperature sensitive sex determination in this species. However, a key caveat to this is that extensive field and laboratory work has previously shown a lack of temperature effects on sex ratios at high elevation compared to a strong effect at low elevation [26, 29]. How do we explain these seemingly contrasting results? One explanation is that sex reversals are always possible at high elevation, but they are rarely expressed because they are masked by female basking behaviour. High elevation females show a greater propensity to bask both in the laboratory and in the wild than do low elevation females [50]. This may allow females to compensate for the poor thermal conditions which greatly reduce, or even eliminate the production of XX male reversal at high elevation. While intuitively sensible this cannot be the sole explanation given, i) we typically observe consistent effects on other thermally sensitive traits (e.g., gestation length) in the high elevation population even in the absence of thermal effects on sex ratio [26] and ii) we observed thermal effects on sex reversals in high elevation females in our thermoregulation treatment despite strong female compensation (Hill et al. unpublished data). Clearly more work is needed to unpick the nature of the relationship between temperature, sex reversals and sex ratio variation if we are to explain how these effects play out across climatic regimes. One key way forward would be to extensively genotype cohorts from our long-term data on *C. ocellatus* for which we have known phenotypic sex ratios as this would show whether (and how much) sex reversal is occurring in the wild in both populations and allow tracking of the prevalence of XX males across seasons. These sample sets are much larger than the numbers achieved here and would therefore have greater power to tease apart the subtleties of these effects. Our observation of XX male offspring from both populations under experimental conditions suggests that the mechanism responsible for geographic divergence in sex ratio responses to climate in *C. ocellatus* is more complex than can be explained by a single process such as sex reversal and that their divergence in sex determination is at an early stage of its evolution. This is consistent with evidence from population genetic data which suggests that the high and low elevation populations of *C. ocellatus* diverged between 0.61 and 0.92 million years ago [51].”

3. Lines 105-107. The author simply state that sex-specific mortality is unlikely to explain sex ratio biases in the lower elevation population. Perhaps state why explicitly?

Response: We appreciate this feedback and have now included a more thorough explanation. We have now included at L95: “Of the mechanisms that could explain GSD+EE at low elevation, sex-specific mortality is unlikely because palpation of gravid females reveals the number of ovulated eggs is equal to the number of offspring [22, 29], which leaves sex reversal as the primary candidate mechanism [6, 21].”

4. Is it safe to assume that males father complete clutches (many of my comments assume this)? Or is it possible to have multiple fathers per clutch?

Response: Thank you for pointing out that this piece of information is missing from the manuscript. High levels of multiple paternity have been observed in this species [11] and we now discuss the potential contribution of multiple paternity to genetic sex ratios and litter sex ratios at L286 “Multiple paternity in this species [46] and other snow skinks [47] means that it is unlikely that entire litters will be fathered by an XX male, however, XX males mating in the population would explain the genetic sex ratios in favour of the XX genotype observed at low elevation.”

5. Lines 24-25. The intraspecific variation in sex ratio is referring to GSD and ESD or two different GSDs? I assumed ESD and GSD the way it is written in the abstract, but in lines 184-186 it sounds like this species has 2 different GSDs? I honestly can't tell for sure. This needs to be made more explicit, and if two different GSDs, then more detail on both sex-determining systems and their distribution would be helpful.

Response: Thank you for highlighting this omission, we agree this is an important point to cover and have now done so at several places throughout the manuscript. The Abstract at L21: “. Here we used thermal experiments to test whether sex reversal is responsible for differences in sex determination in a viviparous reptile, *Carinascincus ocellatus*, a species with XY sex chromosomes and population-specific sex ratio response to temperature.”

In the introduction beginning at L40: “In several vertebrate systems however, the sex encoded genetically can be over-ridden by the environment [2], termed GSD plus environmental effects (GSD+EE; [3]). Mechanistic models suggest that this occurs when the expression of a gene that determines sex is temperature sensitive [4, 5]. Biased sex ratios can result when the homogametic sex (XX females or ZZ males) does not attain the encoded phenotype because of the temperature sensitivity of the sex determining gene product, despite having two copies of that gene. When this occurs, development is diverted down an alternate pathway [4], resulting in individuals with a sex genotype-phenotype mismatch. This outcome is known as ‘sex reversal’ [2, 6] and has been documented in fish [7, 8], amphibians [9] and oviparous reptiles [4, 10, 11]. In amphibians and fish, both the homogametic and heterogametic sex can develop into the opposite phenotype [12, 13].”

In addition, at L71 to L90 we have included the following background on this study system which we feel explains the different sex determination systems in *C. ocellatus* “Field-based, laboratory and theoretical work on *C. ocellatus* has identified intra-specific divergence in sex determination [22] which is rarely observed in amniotes, although reported in amphibians [23] and fish [24, 25]. Long-term field and laboratory data show that sex ratios in a high elevation population do not deviate from parity irrespective of seasonal/developmental temperature [22, 26]. In contrast, in a low elevation population sex ratios correlate with temperature [22]; where sex ratios are female biased in warm seasons/developmental

conditions and male biased in cool seasons/developmental conditions [26, 27]. The population-specific sex ratio response to temperature observed in *C. ocellatus* in the wild has been replicated by manipulating maternal thermal opportunity in the laboratory [22, 26-30]. Low elevation females subjected to cooler short basking treatments (e.g., 4 hours) produce male biased cohorts and those subjected to warmer long basking exposures (e.g., 10 hours) produce female biased cohorts [29, 30]. In contrast, high elevation females subjected to the same thermal regimes produce sex parity cohorts [22, 26-30]. An adaptive explanation for these patterns is that the production of males or females is favoured at different temperatures at low elevation because the concomitant variation in birth date has sex-specific fitness consequences. Warm developmental conditions, and thus early birth, favour females at low elevation because birthdate strongly predicts the onset of maturity and thus reproductive output for females, but not males [22]. The shorter reproductive season at high elevation together with the high seasonal temperature fluctuations mean there are no benefits of early birth to either sex, and therefore, sex ratios remain balanced.”

6. Lines 373-402. The authors describe here how an XX male mating with an XX female could produce both male- and female-biased sex ratios in the absence of sex-reversed (XY) females. It seems to me that this could go a long way to explaining their results. Do the authors find XY males and XX males in the same clutch? Because if not, then wouldn't this suggest the father of the clutch is an XX male? More generally, it seems like including some information on the clutch-level response to temperature would shed some light on the findings presented.

Response: Thank you for highlighting the need to include this information. There is a high incidence of multiple paternity in this species (approximately 94% of clutches have multiple fathers [11]) and it is unlikely that entire clutches have the same father. We now report the occurrence of XX males, XY males and XX females in the same litter: at L202 of the results section: “We found XX and XY males and XX females within the same litter. Specifically, of the 19 and 10 litters containing XX males from low and high elevation respectively, a total of five litters also contained XX females and XY males (two from low and three from high elevation respectively).”

We discuss the relevance of multiple paternity to population sex ratios:

at L283 “Multiple paternity in this species [46] and other snow skinks [47] means that it is unlikely that entire litters will be fathered by an XX male, however, XX males mating in the population would explain the genetic sex ratios in favour of the XX genotype observed at low elevation.”

Journal Name: Proceedings of the Royal Society B
Journal Code: RSPB
Print ISSN: 0962-8452
Online ISSN: 1471-2954
Journal Admin Email: proceedingsb@royalsociety.org
MS Reference Number: RSPB-2021-1778
Article Status: REJECTED

References

1. Holleley, C.E., O'Meally, D., Sarre, S.D., Marshall-Graves, J.A., Ezaz, T., Matsubara, K., Azad, B., Zhang, X., and Georges, A., 2015. Sex reversal triggers the rapid transition from genetic to temperature-dependent sex. *Nature*. **523**: p. 79-82.
2. Quinn, A.E., Sarre, S.D., Ezaz, T., Marshall Graves, J.A., and Georges, A., 2011. Evolutionary transitions between mechanisms of sex determination in vertebrates. *Biol Lett*. **7**(3): p. 443-8.
3. Sarre, S.D., Ezaz, T., and Georges, A., 2011. Transitions between sex-determining systems in reptiles and amphibians. *Annu Rev Genomics Hum Genet*. **12**: p. 391-406.
4. Ezaz, T., Sarre, S.D., O'Meally, D., Graves, J.A.M., and Georges, A., 2009. Sex chromosome evolution in lizards: independent origins and rapid transitions. *Cytogenetic and Genome Research*. **127**(2-4): p. 249-260.
5. Holleley, C.E., Sarre, S.D., O'Meally, D., and Georges, A., 2016. Sex reversal in reptiles: reproductive oddity or powerful driver of evolutionary change? *Sex Dev*. **10**(5-6): p. 279-287.
6. Wapstra, E., Olsson, M., Shine, R., Edwards, A., Swain, R., and Joss, J.M., 2004. Maternal basking behaviour determines offspring sex in a viviparous reptile. *Proc Biol Sci*. **271**: p. S230-232.
7. Cadby, C.D., Jones, S.M., and Wapstra, E., 2014. Geographical differences in maternal basking behaviour and offspring growth rate in a climatically widespread viviparous reptile. *Journal of Experimental Biology*. **217**(7): p. 1175-1179.
8. Caldwell, A.J., While, G.M., and Wapstra, E., 2017. Plasticity of thermoregulatory behaviour in response to the thermal environment by widespread and alpine reptile species. *Animal Behaviour*. **132**: p. 217-227.
9. Wapstra, E., Uller, T., Sinn, D.L., Olsson, M., Mazurek, K., Joss, J., and Shine, R., 2009. Climate effects on offspring sex ratio in a viviparous lizard. *Journal of Animal Ecology*. **78**(1): p. 84-90.
10. Pen, I., Uller, T., Feldmeyer, B., Harts, A., While, G.M., and Wapstra, E., 2010. Climate-driven population divergence in sex-determining systems. *Nature*. **468**(7322): p. 436-438.
11. Uller, T. and Olsson, M., 2008. Multiple paternity in reptiles: patterns and processes. *Mol Ecol*. **17**(11): p. 2566-80.

Appendix B

12th May 2022

Dear Dr Neiman,

We would like to express our appreciation to yourself and the referee for reviewing our resubmission of our manuscript “Sex reversal explains some, but not all, climate mediated sex ratio variation within a viviparous reptile.” We have made the change required and are excited that this work will be published in *Proceedings of the Royal Society B*.

Our response is embedded in the text below highlighted in this colour.

Best regards,

Peta Hill on behalf of all co-authors.

Referee: 3

Comments to the Author(s).

I believe the authors have sufficiently addressed all the comments by the reviewers and incorporated many of the suggestions into the manuscript. I do not have any concerns, and I believe the manuscript would be a great contribution to the journal.

We thank referee 3 for their time and effort in assessing the resubmission of our manuscript. Our response is below.

Note: in Table 4, the numbers for the XX genotypes do not add up to 114, but to 113. Comparing the numbers with the previous tables, I suspect that there is a typo in the number of XX females (should possibly be 95 instead of 94).

Response: Thank you for drawing our attention to this error. We have reviewed our data and amended the total number of XX genotypes in the low elevation population to 113. We note that this also changed the output from the Pearson’s \$\chi^2\$ test, therefore we amended these statistics also. Table 4 as it now appears is below.

Table 4. Cohort genetic sex ratios for offspring from low and high elevation populations of *Carinascincus ocellatus* females held in ‘thermoregulation’ (Long 10h, Short 4h) and ‘no thermoregulation’ (High 33.0 °C, Med 29.5 °C, Low 26.0 °C) experiments. Summary statistics from Pearson’s χ^2 test of the deviation of the genetic sex ratios from parity are included.

Population	Male genotypes		Female genotypes		Genetic sex ratio	Pearson’s χ^2 test
	XY	XX	XY	XX	XY:XX	
Low elevation	80	19	0	94	80:113	$\chi^2 = 5.6$, $p = 0.02$
High elevation	119	12	0	98	119:110	$\chi^2 = 0.35$, $p = 0.55$